# Quantifying the Effects of Combination Trastuzumab and Radiation Therapy in Human Epidermal Growth Factor Receptor 2-Positive Breast Cancer

**DOI:** 10.3390/cancers14174234

**Published:** 2022-08-31

**Authors:** Meghan J. Bloom, Patrick N. Song, John Virostko, Thomas E. Yankeelov, Anna G. Sorace

**Affiliations:** 1Department of Biomedical Engineering, The University of Texas, Austin, TX 78712, USA; 2Department of Radiology, The University of Alabama, Birmingham, AL 35294, USA; 3LiveSTRONG Cancer Institutes, The University of Texas, Austin, TX 78713, USA; 4Department of Oncology, The University of Texas Dell Medical School, Austin, TX 78701, USA; 5Department of Diagnostic Medicine, The University of Texas, Austin, TX 78712, USA; 6Oden Institute for Computational and Engineering Sciences, The University of Texas, Austin, TX 78712, USA; 7Department of Imaging Physics, MD Anderson Cancer Center, Houston, TX 77030, USA; 8Department of Biomedical Engineering, The University of Alabama, Birmingham, AL 35294, USA; 9O’Neal Comprehensive Cancer Center, The University of Alabama, Birmingham, AL 35233, USA

**Keywords:** HER2+, Herceptin, radiosensitive, BT474

## Abstract

**Simple Summary:**

In vitro, combination trastuzumab and single-dose radiotherapy induces an additive effect. In vivo, combination trastuzumab and single-dose radiotherapy induces greater tumor kill than a higher single dose of radiotherapy, suggesting that combination therapy can be considered to achieve a similar reduction in tumor burden as traditional radiotherapy, with fewer adverse effects.

**Abstract:**

Background: Trastuzumab induces cell cycle arrest in HER2-overexpressing cells and demonstrates potential in radiosensitizing cancer cells. The purpose of this study is to quantify combination trastuzumab and radiotherapy to determine their synergy. Methods: In vitro, HER2+ cancer cells were treated with trastuzumab, radiation, or their combination, and imaged to evaluate treatment kinetics. In vivo, HER2+ tumor-bearing mice were treated with trastuzumab and radiation, and assessed longitudinally. An additional cohort was treated and sacrificed to quantify CD45, CD31, α-SMA, and hypoxia. Results: The interaction index revealed the additive effects of trastuzumab and radiation in vitro in HER2+ cell lines. Furthermore, the results revealed significant differences in tumor response when treated with radiation (*p* < 0.001); however, no difference was seen in the combination groups when trastuzumab was added to radiotherapy (*p* = 0.56). Histology revealed increases in CD45 staining in tumors receiving trastuzumab (*p* < 0.05), indicating potential increases in immune infiltration. Conclusions: The in vitro results showed the additive effect of combination trastuzumab and radiotherapy. The in vivo results showed the potential to achieve similar efficacy of radiotherapy with a reduced dose when combined with trastuzumab. If trastuzumab and low-dose radiotherapy induce greater tumor kill than a higher dose of radiotherapy, combination therapy can achieve a similar reduction in tumor burden.

## 1. Introduction

Human epidermal growth factor receptor 2-positive (HER2+) breast cancer is characterized by the overexpression of the HER2/neu receptor and represents 15–20% of annual breast cancer cases [1]. Trastuzumab, a monoclonal antibody targeted to the HER2 receptor, is used clinically in both neoadjuvant and adjuvant standard-of-care treatment, and is often combined with radiation in the adjuvant setting to eliminate residual disease and prevent recurrence [2,3,4]. Preclinical studies have shown that the overexpression of HER2 is a contributing factor to radiation resistance, and treatment with anti-HER2 therapy could potentially act as a radiosensitizer [5,6,7,8,9]. However, HER2+ patients still have a greater chance of recurrence than patients with HER2- status [10,11]. Thus, there is a need to investigate the optimization of trastuzumab and radiation in a combination treatment regimen. 

One mechanism by which HER2 overexpression promotes tumorigenesis is through the inhibition of apoptosis by the upregulation of the phosphatidylinositol 3’-kinase–protein kinase B–mammalian target of rapamycin (PI3K/AKT/mTOR) pathway. Trastuzumab blocks PI3K/AKT/mTOR pathway signaling, causing cell cycle arrest and inducing cancer cell death [12,13]. Studies by Guo et al. and Liang et al. found that, when used in combination, trastuzumab sensitizes HER2+ breast cancer cells to radiation therapy in vitro, and that the upregulation of the PI3K/AKT pathway may be specifically involved in radioresistance in HER2+ breast cancer [5,7]. Additionally, breast tumors can become hypoxic due to the irregular and tortuous vasculature, which increases radioresistance [14,15,16]. Previous results in HER2+ breast cancer revealed that treatment with trastuzumab could increase tumor oxygenation in a pre-clinical mouse model through the stabilization of tumor blood vessels [17]. Although there has been investigation into the potential ability of trastuzumab to radiosensitize breast cancer, studies lack longitudinal data quantifying sensitization over time, reducing its clinical relevance. Furthermore, the effects of the order of therapies have not been systematically investigated [18]. 

Common in vitro approaches for evaluating cell responses to radiation therapy include end-point analyses, such as the 3-[4,5-dimethylthiazolyl-2]-2,5-diphenyltetrazolium bromide (MTT) assay or the clonogenic assay. Furthermore, the results evaluating cell viability after treatment can largely vary depending on the end time-point chosen [19]. Additionally, these methods lack the ability to longitudinally observe cell response, which is critical in determining underlying changes in cell physiology during therapy [20]. 

This contribution has two goals: (1) to investigate and quantify the effects of radiation and trastuzumab longitudinally in vitro, and (2) to test the hypothesis that trastuzumab sensitizes HER2+ breast cancer to radiation therapy in an in vivo model of HER2+ breast cancer. This study uses the IncuCyte Live-Cell Analysis System (Essen Bioscience, Ann Arbor, MI, USA), which delivers real-time data of cellular responses and has been shown to be accurate in analyzing changes in breast cancer cell proliferation over time [21]. Quantifying the synergy between trastuzumab and radiation over time has the potential to elucidate optimal combination regimens and decrease the amount of therapy needed to achieve tumor control or eradication. 

## 2. Materials and Methods

### 2.1. Cell Culture

BT474, SKBR3, and MDA-MB-231 cell lines were acquired from ATCC. BT474 cells were maintained in improved minimal essential medium (IMEM, Invitrogen, Carlsbad, CA, USA) supplemented with 10% FBS, 1% penicillin/streptomycin, and 20 μg/mL of insulin. SKBR3 cells were cultured in McCoy’s 5A medium (ATCC, Manassas, VA, USA) supplemented with 10% FBS and 1% penicillin/streptomycin. MDA-MB-231 cells were maintained in Dulbecco’s Modified Eagle’s Medium (Thermo Fisher Scientific Inc., Waltham, MA, USA) supplemented with 5% FBS and 200 μg/mL of G418. All cells were grown at 37 °C with 5% CO_2_. The cells were cultured to 70–80% confluency and the cell counts were determined with a Countess II FL automated cell counter (Thermo Fisher Scientific Inc., Waltham, MA, USA). All used cell lines were acquired from ATCC. 

### 2.2. Western Blot Evaluation of HER2 Expression and Quantification

BT474, SKBR3, and MDA-MB-231 breast cancer cells were washed with ice-cold phosphate-buffered saline and lysed for 10 min on ice with radioimmunoprecipitation assay (RIPA) buffer supplemented with protease inhibitor (Roche Applied Science, Indianapolis, IN, USA). The lysates were centrifuged at 13,400× *g* for 20 min at 4 °C and collected to undergo the bicinchoninic acid (BCA) protein quantification assay using a Nanodrop 2000 c spectrophotometer (Thermo Fisher Scientific, Waltham, MA, USA). For each cell line, 20 μg of protein was prepared in a solution of sodium dodecyl sulfate (SDS) and β-mercaptoethanol, and boiled for 5 min at 95° C. The samples were run on NuPAGE Bis-Tris gel and transferred to a polyvinylidene fluoride (PVDF) membrane. The membrane was blocked in 5% dry milk in tris-buffered saline (TBST), probed with horseradish peroxidase (HRP)-conjugated mouse anti-human β-actin overnight at 4 °C, and developed with an Amersham ECL western blot detection system (GE Healthcare, Buckinghamshire, UK) for one minute at room temperature. Membranes were developed in the absence of light and visualized with an SRX-101A Medical Film Processor (Konica Minolta Medical and Graphic, Inc., Shanghai, China). The membrane was incubated in stripping buffer for 15 min, washed with TBST, and probed with Rabbit anti-human HER2/ErbB2 primary antibody (Cell Signaling Technology, Danvers, MA, USA) for 2 h at room temperature. The membrane was washed and incubated with HRP-conjugated goat anti-rabbit IgG secondary antibody (Cell Signaling Technology, Danvers, MA, USA) for 1 h at room temperature. After a final wash, the membrane was redeveloped and visualized for HER2 expression. The quantification of bands was conducted with the Image Studio Lite program, version 5.0 (LI-COR Biosciences, Lincoln, NE, USA). HER2 expression was normalized to β-actin expression for quantification. 

### 2.3. Transfection of Breast Cancer Cell Lines

BT474, SKBR3, and MDA-MB-231 cells were transfected to express enhanced green fluorescent protein (EGFP) through the Sleeping Beauty Transposon System. An EGFP plasmid was obtained and cloned into a Sleeping Beauty-compatible vector, psDBbi-Neo (Addgene, Cambridge, MA, USA plasmid #60525). Lipofectamine LTX (Thermo Fisher Scientific, Waltham, MA, USA) was used to co-transfect the psDBbi-Neo vector and the pCMV (CAT) T7-SB100 (Addgene, Cambridge, MA, USA, plasmid #34879) Sleeping Beauty transposase. Following transfection, cells were cultured in their respective media supplemented with 200 μg/mL G418 for four weeks to select for positively transfected cells. Fluorescence-activated cell sorting was used to separate the cells, and the 25% highest intensity EGFP-expressing cells were used in the experiments. The pCMV (CAT)T7-SB100 plasmid was a gift from Dr. Zsuzsanna Izsvak [22], and the pSBbi-Neo was a gift from Dr. Eric Kowarz [23]. 

### 2.4. In Vitro Treatment Experiments

#### 2.4.1. Experiment 1: Order of Dosing

BT474-GFP cells were plated in 96-well plates at 5000 cells/well. Twenty-four hours later (Day 0), the plates were placed in an IncuCyte S3 Live-Cell Analysis System (Essen BioScience, Ann Arbor, MI, USA) and whole-well imaged every six hours with the phase contrast and green channels (excitation 440–480, emission 504–544) for one week. Single-agent trastuzumab treatment groups were treated with trastuzumab (0.1 ng/mL) from 0 to 48 h (group 1), 24–72 h (group 2), or 48–96 h (group 3). For the single-agent radiation treatment groups, cells were treated with radiation (10 Gy) at 24 h with a CellRad Dedicated Benchtop Cell Irradiator (Faxitron, Tucson, AZ, USA). For the combination treatment groups, the cells received trastuzumab (0.1 ng/mL) before, with, or after radiation treatment (10 Gy), as detailed in Figure 1A. Each treatment group had 24 replicates per experiment. The experiments were performed three times. 

#### 2.4.2. Experiment 2: Quantifying Radiation + Trastuzumab Combination Effects

BT474-GFP, SKBR3-GFP, or MDA-MB-231-GFP cells were plated in 96-well plates at 5000 cells/well. Twenty-four hours later (Day 0), the cells were treated with either 0.1% saline, trastuzumab (0.1 ng/mL), or radiation (5 or 10 Gy) for the control and single-agent treatment groups. Combination groups received both trastuzumab (0.1 ng/mL) and radiation (5 or 10 Gy) on Day 0 (Figure 1B). The plates were placed in an IncuCyte S3 Live-Cell Analysis System (Essen BioScience, Ann Arbor, MI, USA) and whole-well imaged (4×, 2.82 μm/pixel) every six hours with the phase contrast and green channels (excitation 440–480, emission 504–544) for one week. Each treatment group had 30 replicates per experiment. The experiments were performed three times.

### 2.5. Image Analysis

Images were analyzed using an IncuCyte S3 Live-Cell Analysis System (Essen BioScience, Ann Arbor, MI, USA) to segment discrete green fluorescent objects (cell counts). The background signal was estimated in 100 μm segments and subtracted from the image. The edges of clumped cells were determined to be the dimmest point between two objects. Objects greater than 1000 μm^2^ were filtered out of the analysis. Cell growth was normalized to the initial seeding density (0 h).

### 2.6. Interaction Index Calculation

The interaction index [24,25] based on the Bliss independence was used to determine the effects of combined radiation and trastuzumab treatment in vitro. To calculate the effects of the radiation (*A*), trastuzumab (*B*), and combination (*AB*) treatments compared with the control (*C*), let (*f*) represent the difference in normalized cell growth (*N*): (1)$$\;f_{i}=\frac{N_{C}-N_{i}}{N_{C}}\;i=A,\;B,\;AB$$

The interaction index (*I*) can be calculated by Equation (2):(2)$$\;I=f_{AB}-f_{A}-f_{B}+f_{A}f_{B}$$
and the value of *I* determines the effect of the combination therapy: (3)$$\;I=\{\begin{array}{c} Synergistic\;\;\;\;\;\;\;\;I>0\\ \;\;\;\;Additive\;\;\;\;\;\;\;\;\;\;I=0\\ Antagonistic\;\;\;\;\;I<0 \end{array}\;\;\;\;\;\;$$

### 2.7. Animal Procedures

Female nude athymic mice (NU/J) (N = 47) were purchased from The Jackson Laboratory (Bar Harbor, ME, USA) at 3–4 weeks of age with an average weight of 12–15 g and acclimated for one week in housing with a normal light cycle, sterile water and food, and microisolation cages. The mice were subcutaneously implanted with a 0.72 mg, 60-day release, 17β-estradiol pellet (Innovative Research of America, Sarasota, FL, USA) in the nape of the neck. One day later, 10^7^ BT474 cells were implanted subcutaneously in 100 μL of serum-free IMEM media with 30% growth-factor-reduced Matrigel. The tumor volume was calculated using the formula: (4/3) × Π × (transverse tumor diameter/2) × (longitudinal tumor diameter/2) × (average of (transverse × longitudinal tumor diameter/2)). The tumors were grown to approximately 250 mm^3^ (8–10 weeks) and entered into the study. The animals were randomly sorted into treatment groups: group (1) control: 100 μL of saline on Days 0 and 3 (N = 6), group (2) trastuzumab alone: 10 mg/kg on Days 0 and 3 (N = 7), group (3) 5 Gy of radiation alone on Day 0 (N = 6) and 100 μL of saline on Day 3, group (4) 10 Gy of radiation alone on Day 0 (N = 5) and 100 μL of saline on Day 3, group (5) Radiation 5 Gy on Day 0 + trastuzumab 10 mg/kg on Days 0, 3 (N = 5), group (6) 10 Gy of radiation on Day 0 and 10 mg/kg of trastuzumab on Days 0 and 3 (N = 5). trastuzumab and saline were administered via intraperitoneal (IP) injection. Radiation treatments (~4 Gy/min, 225 kV, 17.8 mA) were administered using a MultiRad 350 Irradiation System (Faxitron, Tucson, AZ, USA). Tumor measurements were taken three times per week using calipers for four weeks, at which point the mice were humanely euthanized via sustained isofluorane exposure (5% isofluorane for 5 min), followed by cervical dislocation. Mice in the immunohistochemistry cohort were divided into three treatment groups: (1) 5 Gy of radiation on Day 0 (N = 4) and 100 μL of saline on Day 3, (2) 10 mg/kg of trastuzumab on Days 0 and 3 (N = 4), and (3) 5 Gy of radiation on Day 0 and 10 mg/kg of trastuzumab on Days 0 and 3 (N = 5). Tumor growth was measured three times with calipers for one week, at which point the mice were euthanized and tumors were extracted. One hour prior to sacrifice, the mice were intravenously injected with 60 mg/kg of pimonidazole (Hypoxyprobe, Inc., Burlington, MA, USA) in 100 μL of saline. Tumors were cut across the longest cross-section and half was fixed in 10% neutral-buffered formalin (Fisher Scientific International Inc., Pittsburgh, PA, USA) for 48 h and then placed in 70% ethanol. The normalized tumor volume was calculated with the formula: ((*X*_1_ − *X*_0_)/*X*_0_) × 100, where *X*_0_ and *X*_1_ represent the tumor volume at baseline and tumor volume at subsequent timepoints, respectively.

### 2.8. Immunohistochemistry 

Formalin-fixed tumor sections were embedded in paraffin and sliced into four-micron sections. The sections were stained for hematoxylin and eosin (H&E), mouse anti-CD31, mouse anti-α-smooth muscle actin (α-SMA, Abcam, Cambridge, UK), mouse anti-CD45 (Invitrogen, Carlsbad, CA, USA), and anti-pimonidazole (Hypoxyprobe, Inc., Burlington, MA, USA). The immuno-stained slides were scanned (20×, 0.495 μm/pixel) with an Aperio ScanScope (Leica Microsystems, Wetzlar, Germany). Regions of necrosis were determined using manual segmentation. All other stains were quantitatively segmented based on the color thresholds determined from positive and negative controls following in-house MATLAB (MathWorks Inc., Natick, MA, USA) routines. The images were converted to grayscale for registration to the corresponding tumor H&E. Then, transformations consisting of translation, rotation, and scale (similarity) based on intensity were applied. A viable tissue mask was defined as the total tumor area minus the necrotic area. Inflammation (CD45), hypoxia (pimonidazole), and apoptosis (caspase3) were defined as the percent of positive stain per viable tissue area. The microvessel density (CD31) and vascular smooth muscle coverage (α-SMA) were calculated as the number of vessels per square millimeter of tumor tissue. The vessel maturation index was evaluated as the ratio of α-SMA coverage to microvessel density. 

### 2.9. Statistical Analysis

Statistical analysis was conducted using MATLAB (MathWorks Inc., Natick, MA, USA). A two-way analysis of variance (ANOVA), adjusting for multiple comparisons with Dunn’s, was used to determine longitudinal differences in the order of dosing experiments. All in vitro data are presented as the mean ±95% confidence interval, with *p* < 0.05 indicating significance. Statistical differences between in vivo tumor growth at each time point and ex vivo immunohistochemistry samples were determined using a nonparametric Wilcoxon rank sum test. All in vivo and ex vivo data are presented as the mean ± standard error, with *p* < 0.05 indicating significance. The Friedman’s test adjusting for multiple comparisons with Dunn’s was used to assess longitudinal changes in tumor growth between treatment groups.

## 3. Results

### 3.1. HER2 Expression in BT474, SKBR3, and MDA-MB-231 Cell Lines

The quantification of HER2 protein expression in BT474, SKBR3, and MDA-MB-231 cells confirmed the reported HER2 status in each cell line (Figure 2). The Western blot results revealed visible bands of HER2 protein in the BT474 and SKBR3 HER2+ cell lines at 185 kDa, and showed little to no expression in MDA-MB-231 cells, a HER2- cell line (Figure 2A). β-actin was used as an internal control for the normalization of HER2 expression. The ratios of HER2:β-actin in BT474 and SKBR3 cells were 1.41 and 1.46, respectively, confirming HER2 overexpression. MDA-MB-231 cells had a HER2:β-actin ratio of 0.08, confirming no HER2 overexpression (Figure 2B). 

### 3.2. Order of Radiation and Trastuzumab Therapy Does Not Affect Cell Death Response In Vitro

Cells were administered with trastuzumab alone as a single agent control from either 0–48 h, 24–72 h, or 48–96 h. No significant differences were observed between these groups (Figure 3). In combination regimens, cells were treated with trastuzumab (0.1 ng/mL) 24 h before radiation (10 Gy) treatment, at the same time as radiation treatment, and 24 h after radiation treatment. No significant differences were observed in cell growth over time when comparing trastuzumab treatment before with treatment at the same time (*p* = 1.00), or treatment after (*p* = 0.98). Additionally, treatment with trastuzumab at the same time as radiation did not alter the cell response over time compared with treatment after radiation (*p* = 0.93) (Figure 3). As the order of dosing did not affect the cell response to therapy, further in vitro experiments were conducted administering trastuzumab and radiation at the same time (Figure 1B). 

### 3.3. Quantification of Longitudinal Cell Growth after Combination Therapy Reveals Additive Effects In Vitro

Figure 4 displays BT474, SKBR3, and MDA-MB-231 cell proliferation after being treated with trastuzumab (0.01 ng/mL), radiation (5 or 10 Gy), or combination treatment over one week. No significant differences were observed over time in MDA-MB-231 cell proliferation between the control and trastuzumab single-agent treatment groups (*p* = 0.88, panels e and f of Figure 4), which is to be expected in a HER2- cell line not responsive to trastuzumab. There were also no significant differences in proliferation between the radiation single-agent or combination treatment groups as expected (*p* = 0.84, Figure 4E and *p* = 0.80, Figure 4F). Figure 5 displays the interaction index over time for each treatment group in Figure 4. No group had an interaction index that fell significantly above or below 0 at any time point (*p* > 0.05), indicating additive treatment effects. 

### 3.4. Tumor Size Decreases Faster with Combination Therapy than Either Single Therapy In Vivo 

Figure 6A displays the tumor volume changes in response to trastuzumab, radiation, and combination regimens over four weeks. Mice treated with 10 Gy radiation had a significantly greater decrease in tumor size than mice treated with 5 Gy radiation over time (*p* < 0.001) (Figure 6B). However, mice treated with trastuzumab + 5 Gy radiation revealed no statistical differences in tumor response over time compared with mice treated with trastuzumab and 10 Gy radiation (*p* = 0.56) (Figure 6C). Additionally, mice treated with trastuzumab and 5 Gy radiation had significantly smaller tumors than mice treated with trastuzumab alone from Day 7 onwards (*p* < 0.05), with the exception of Day 9 (*p* = 0.11). Mice treated with trastuzumab and 5 Gy radiation had significantly smaller tumors than mice treated with 5 Gy alone from Day 2 onwards (*p* < 0.05), with the exception of Day 25 (*p* = 0.25). Mice treated with trastuzumab and 5 Gy radiation had significantly smaller tumors than mice treated with 10 Gy radiation on Days 2–17 (*p* < 0.05). 

### 3.5. Tumor Immune Infiltration Was Higher in Mice That Received Trastuzumab Treatment 

Figure 7A displays representative images of CD45+ staining in mice treated with radiation (5 Gy) on Day 0, trastuzumab (10 mg/kg) on Days 0 and 3, or combined radiation and trastuzumab. Mice treated with trastuzumab alone had significantly higher CD45+ staining (2.63 ± 0.73%) than mice treated with radiation alone (0.77 ± 0.13%) (*p* = 0.03) on Day 7. Mice treated with combination therapy also had significantly higher CD45+ staining (2.48 ± 0.44%) than mice treated with radiation alone (*p* = 0.03). No significant differences were observed between mice treated with trastuzumab alone and the combination treatment (*p* = 1.00) (Figure 7B). No significant differences were observed between the treatment groups (*p* > 0.05) in terms of the percent pimonidazole (Appendix A) or vascular maturation index (Appendix A). 

## 4. Discussion

Trastuzumab and radiation are commonly used as adjuvant therapies in the clinical treatment of HER2+ breast cancer. Radiation therapy can reduce recurrence rates in patients by eradicating the remaining cancer cells after the surgical removal of tumors; however, HER2+ patients have greater chances of recurrence than HER2- patients [11,26]. In vitro evidence suggests that trastuzumab can sensitize HER2+ breast cancer cells to radiation through the inhibition of signaling pathways involved in DNA repair mechanisms [7,27,28], although, systematic evaluations of the order of therapies and longitudinal data of cell responses are limited. This study demonstrates that the order of therapy does not alter cell proliferation after treatment, and longitudinal treatment effects are additive in vitro. Furthermore, in vivo, tumors regress faster when treated with combination therapy than either single-agent therapy, and there is evidence that immune modulation may impact the treatment response. 

The interaction index calculations revealed additive effects of combination radiation and trastuzumab treatment across all HER2+ breast cancer cells evaluated. This contrasts with the findings of others who have evaluated cell responses to combination radiation and trastuzumab using end-point assays [7,9]. For example, Liang et al. found a synergistic effect of trastuzumab and radiation treatment with BT474 and SKBR3 cells 24 h after therapy using an ELISA to evaluate apoptosis (cytoplasmic histone-associated DNA fragments) [7]. In their study, cells were pre-incubated with trastuzumab for 16 h before radiation treatment; however, our results did not indicate that pre-treatment with trastuzumab altered cell death under radiation. Rao et al. found a similar synergistic effect using SUM-149PT cells treated with a HER2 tyrosine–kinase inhibitor in combination with radiation therapy; however, this study used a longer pre-incubation time with trastuzumab (7 days) and evaluated cell growth two weeks after radiation treatment using a clonogenic assay [9]. The differences in the results could have arisen from the alternate methods used to detect cellular responses to treatment, different dosing strategies, and differences in the timing of evaluation after treatment. One important aspect of our study is that we have quantified longitudinal cell growth after variations in combination treatments, thereby providing a better understanding of the cellular dynamics after combination treatment compared with traditional end-point assays. This provides the opportunity to assess the long-term effects of the order of sequencing of therapies and variations in dosing strategies. 

In vivo tumor growth data show significant differences in the longitudinal tumor response between groups that were treated with 5 Gy and 10 Gy of radiation; however, no significant difference in the longitudinal tumor response was observed between the doses of radiation when trastuzumab was added to the treatment regimen. This finding suggests that a smaller dose of radiation could be used in combination with trastuzumab without decreasing efficacy. Additionally, the results show a faster rate of tumor regression in groups treated with combination therapy compared with either single therapy. The results of this research provide a framework to examine treatment interactions between radiation therapy and targeted treatments, and can be used to guide combination therapies to enhance treatment efficacy while minimizing off-target side effects. This approach will be particularly important, as there are increased individualized medicine plans and de-escalated radiation treatments. 

To evaluate if microenvironmental alterations could be impacting treatment responses, immune infiltration, hypoxia, and the vascular maturation index were evaluated in tumors treated with 5 Gy radiation, trastuzumab, and 5 Gy radiation in combination with trastuzumab seven days after treatment. CD45 histology staining revealed an increase in immune infiltration in treatment groups receiving trastuzumab. Previously collected data revealed that trastuzumab has the potential to reprogram the immunosuppressive components of the tumor microenvironment and could contribute to anti-tumor responses. The evaluation of immune cell characterization and impact on tumor regression in this treatment regimen is needed to further elucidate the mechanisms contributing to enhanced tumor regression in the combination treatment group [29]. Although no significant differences were shown in hypoxia and the vascular maturation index, previous studies using longitudinal imaging have revealed that trastuzumab can reduce hypoxia and increase vascular perfusion [17,30]. It is possible that, by employing longitudinal imaging methods, windows of heightened oxygenation could be revealed, and an alternate order of dosing could enhance the treatment response in vivo. 

The limitations of this study include the lack of longitudinal data evaluating microenvironmental changes corresponding with tumor growth dynamics. Immuno–PET imaging is an alternative method that could longitudinally evaluate immune infiltration after treatment [31]. ^18^F-Fluoromisonidazole positron emission tomography is one method used to monitor the tumor oxygenation levels, and dynamic contrast-enhanced magnetic resonance imaging is an alternative method to quantitatively assess vascularity [27,30]. Neither of these methods would allow for histological validation at every time point; however, they would offer finely time-resolved data of tumor status over the course of treatment. This study used a single dose of radiation and provided preliminary data showing that a lesser dose of radiation could be used in combination with trastuzumab to achieve the same therapeutic efficacy as a higher dose of radiation. Further studies should be conducted to evaluate efficacy with a clinical treatment regimen that uses fractionated dosing. Additional trastuzumab treatment doses were investigated for the in vitro experiments in this study. Doses higher than 0.01 ng/mL exhibited greater decreases in cell proliferation in the single-agent treatment groups; however, they did not affect the combination treatment response. Doses lower than 0.01 ng/mL had no significant effect on HER2+ cell growth, which is not biologically observed in vivo; therefore, a sub-therapeutic dose of trastuzumab (0.01 ng/mL) was chosen as the appropriate treatment condition to assess synergy. It is possible that using a higher seeding density would allow for the evaluation of cell proliferation with a higher trastuzumab dose, although, when cells become confluent, the IncuCyte system is less accurate in quantifying cell numbers [21]. 

## 5. Conclusions

This study demonstrates that the longitudinal assessment of growth dynamics can result in different conclusions than end-point assays, and differences in methodologies should be taken into consideration when evaluating cellular responses to treatment. Furthermore, this study yielded preliminary data that using trastuzumab in combination with radiation has the potential to decrease the dose of radiation required without affecting therapeutic efficacy. Histology data revealed that treatment with trastuzumab increased immune infiltration compared with radiation alone. Further assessment of immune changes in single-agent and combination regimens could identify mechanistic explanations for the differences in therapeutic efficacies. 

## Figures and Tables

**Figure 1 cancers-14-04234-f001:**
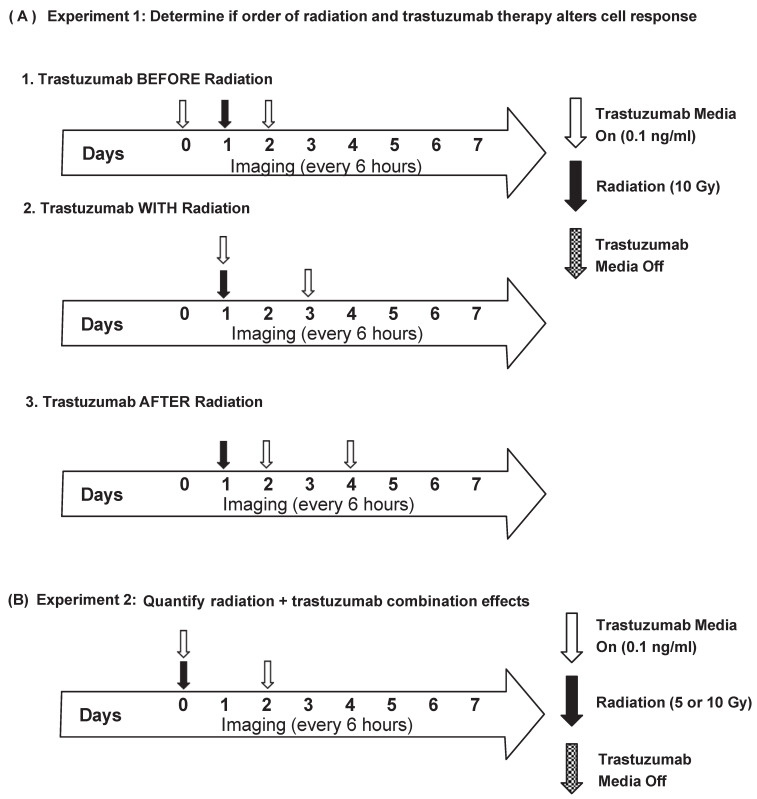
Treatment plans for in vitro experiments. (**A**) Cells were treated with either trastuzumab (0.1 ng/mL) from 0–48 h (1, before radiation), 24–72 h (2, with radiation), or 48–96 h (3, after radiation). All groups received radiation (10 Gy) at 24 h. All groups were imaged every 6 h for seven days. (**B**) Cells were treated with trastuzumab (0.1 ng/mL) and radiation (5 or 10 Gy) at the start of the experiment (Day 0). Trastuzumab was removed 48 h later. Cells were imaged every six hours for seven days.

**Figure 2 cancers-14-04234-f002:**
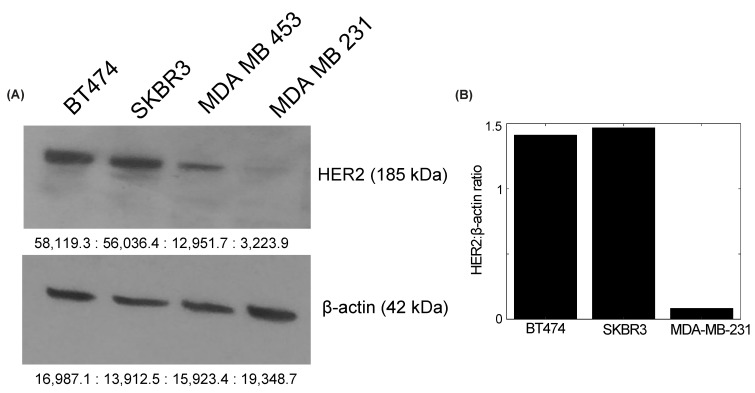
HER2 quantification in BT474, SKBR3, and MDA-MB-231 cells. (**A**) Expression of HER2 protein (185 kDa) in each cell line. β-actin (42 kDa) was used as an internal control. Visible protein bands are seen in the BT474 and SKBR3 HER2+ cell lines, while MDA-MB-231 shows little to no expression. (**B**) HER2:β-actin ratio in each cell line. BT474 and SKBR3 cells have ratios of 1.41 and 1.46, respectively. MDA-MB-231 has a ratio of 0.08, confirming no HER2 amplification. The uncropped blots are shown in Appendix A.

**Figure 3 cancers-14-04234-f003:**
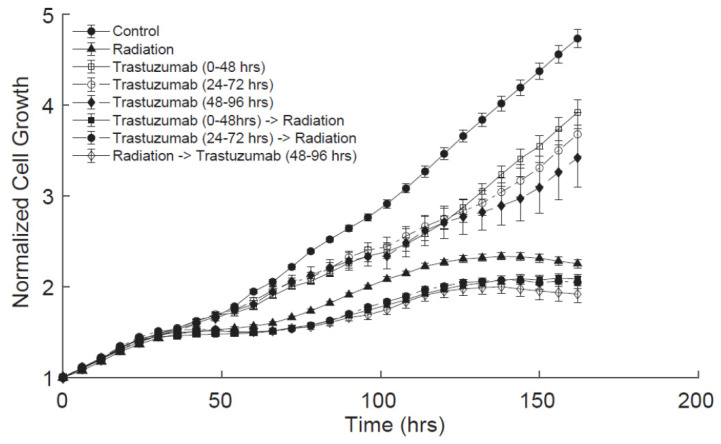
Differences in BT474 cell growth with trastuzumab administered before, at the same time, or after radiation therapy. Treatment with trastuzumab before radiation did not significantly alter cell response to therapy compared with trastuzumab treatment at the same time (*p* = 1.00) or treatment after radiation (*p* = 0.98). Treatment with trastuzumab at the same time did not significantly alter the cell response to therapy compared with treatment after radiation (*p* = 0.93). Altering the order of dosing trastuzumab and radiation did not alter the cell response to treatment in vitro.

**Figure 4 cancers-14-04234-f004:**
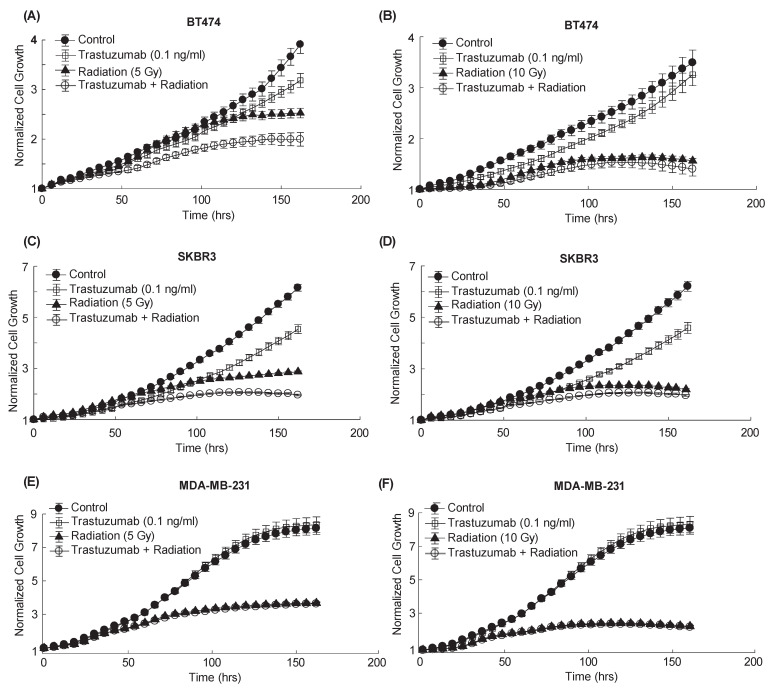
Cell proliferation over one week after radiation, trastuzumab, or combination treatment. All trastuzumab single-agent and combination groups were treated with 0.1 ng/mL of trastuzumab from 0–48 h. All radiation single-agent and combination groups were treated with 5 or 10 Gy radiation on Day 0. Graphs display the proliferation of: (**A**) BT474 cells after trastuzumab, 5 Gy radiation, and combination treatment, (**B**) BT474 cells after trastuzumab, 10 Gy radiation, and combination treatment, (**C**) SKBR3 cells after trastuzumab, 5 Gy radiation, and combination treatment, (**D**) SKBR3 cells after trastuzumab, 10 Gy radiation, and combination treatment, (**E**) MDA-MB-231 cells after trastuzumab, 5 Gy radiation, and combination treatment, and (**F**) MDA-MB-231 cells after trastuzumab, 10 Gy radiation, and combination treatment. No significant difference was observed in the control MDA-MB-231 cell proliferation and cells treated with single-agent trastuzumab (*p* = 0.88, (**E**,**F**)). No significant difference in cell proliferation was observed between cells treated with single-agent radiation and cells treated with combination treatment (*p* = 0.84, (**E**) and *p* = 0.80, (**F**)).

**Figure 5 cancers-14-04234-f005:**
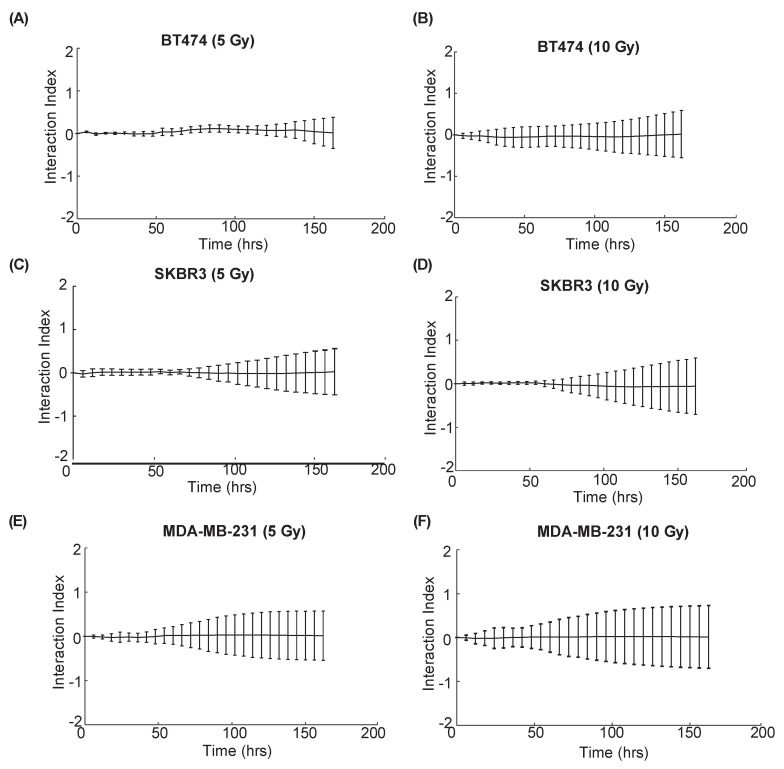
Interaction index calculated per time point over one week of treatment. All groups were treated with 0.01 ng/mL of trastuzumab. Graphs display the interaction indexes of (**A**) BT474 cells with 5 Gy radiation and trastuzumab; (**B**) BT474 cells with 10 Gy radiation and trastuzumab; (**C**) SKBR3 cells with 5 Gy radiation and trastuzumab; (**D**) SKBR3 cells with 10 Gy radiation and trastuzumab; (**E**) MDA-MB-231 cells with 5 Gy radiation and trastuzumab; and (**F**) MDA-MB-231 cells with 10 Gy radiation and trastuzumab. No group had an interaction index significantly above or below 0 at any time point, indicating additive treatment effects.

**Figure 6 cancers-14-04234-f006:**
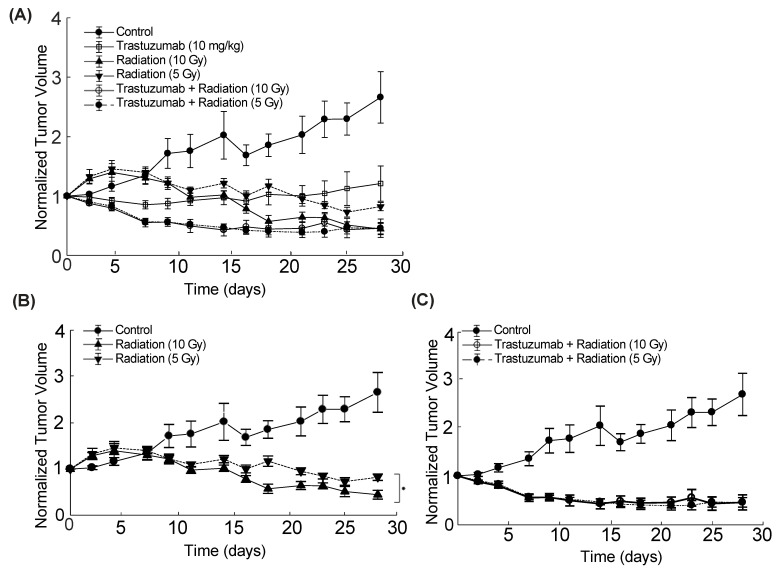
In vivo tumor growth response to single-agent and combination trastuzumab and radiation therapy. All mice were treated with radiation on Day 0 and trastuzumab on Days 0 and 3. (**A**) All groups. (**B**) Control with single-agent radiation therapy. There was a significant difference between groups treated with single-agent 5 Gy and single-agent 10 Gy radiation (* *p* < 0.001). (**C**) Control with trastuzumab and radiation (5 Gy), and trastuzumab and radiation (10 Gy). There was no significant difference between the radiation-treated groups when adding trastuzumab to the regimen (*p* = 0.56).

**Figure 7 cancers-14-04234-f007:**
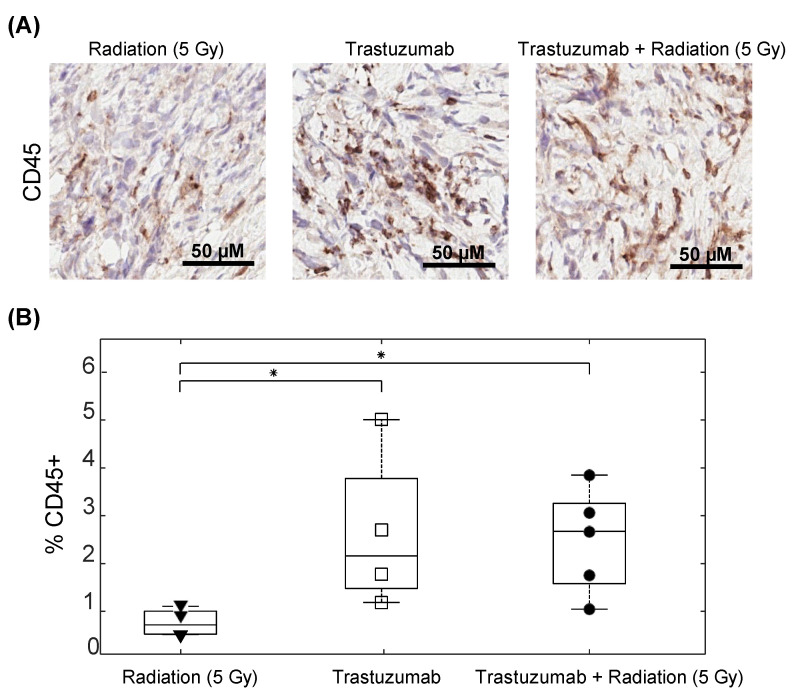
Percent CD45+ in single agent radiation, trastuzumab, and combination-treated tumors on Day 7. All radiation single-agent and combination groups were treated with 5 Gy of radiation on Day 0. All trastuzumab single-agent and combination groups were treated with 10 mg/kg of trastuzumab on Days 0 and 3. (**A**) Representative images of CD45 staining in single-agent and combination treatment groups. (**B**) Percent CD45+ staining from the central slices of tumors in the single-agent and combination treatment groups, revealing significant increases in CD45+ in trastuzumab single-agent and trastuzumab and radiation (5 Gy) treatment compared with radiation alone (* *p* = 0.03). No significant difference was seen in mice treated with single-agent trastuzumab and trastuzumab and radiation (*p* = 1.00).

## Data Availability

The data used in the current study, as well as codes used for analyses, are available from the corresponding author upon reasonable request.

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
