# Peer review of "Quantifying the Effects of Combination Trastuzumab and Radiation Therapy in Human Epidermal Growth Factor Receptor 2-Positive Breast Cancer"

_cancers, 2022, doi:10.3390/cancers14174234_

Round 1

Reviewer 1 Report

The presented manuscript tries to quantify in vitro and in vivo effects of combination trastuzumab and radiation therapy in human epidermal growth factor receptor 2 positive breast cancer. All in all it is a well written manuscript with sound scientific methodology and results that are of interest for the readers of Cancers. The manuscipt is clearly written, relevant for the field and presented in a well-structured manner. The effects of combined trastumab and radiotherapy is an important issue that deserves further research and is of high clinical interest. Therefore this manuscript adds to this research topic. The authors used three cell lines for their research to confirm the results. The references are up to date, the conclusions are supported by the presented data

Some recommendations for minor improvements:

- as a non-native-english speaking reviewer I cannot judge a manuscript written in the USA. But some formal parts of the Abstract appeared a little bit uncommon to me. Please check the language especially of the abstract.

- p5_l181: it seems that there is a formatting problem - please resolve

- Please delete extra spaces at the beginning of the sentences. This appears 3-4 times in the manuscript. This is not common for this journal.

- References: 4/28 (14%) of the references appear to me as previous work of the same group. Please check if this is necessary.

- p12_l358: cells

- p13_l379: "The finding suggests ...", this sentence appears to me very speculative. I my view the presented preclinical data is not applicable to the clinical situation of real patients, it is hyothesis-generating. Please find a more cautious expression regarding the clinical use.  

Reviewer 2 Report

The publication shows very well designed experiments with cell culture, brings very interesting results and most importantly shows safe steps that are useful in clinical work. 

Reviewer 3 Report

The authors are commended for investigating this interesting topic about the synergy of radiation and trastuzumab. The authors have shown an additive effect with the combined treatment that permitted lower doses of radiation with similar outcomes. Despite this study being applied on mice with no direct clinical correlation, it still serves as a good insight to clinical studies as we move towards an era of individualized medicine and de-escalated radiation treatments.

The authors explained their methodology clearly and the paper was well-organized and adequately concise with acceptable English language use. The tables and figures were helpful and clear. The statistical methods were well-explained and the statistical analysis was appropriate to the project as described.
